# CG-MLLM: Captioning and Generating 3D Content via Multi-modal Large Language Models

Junming Huang [1 2]   Chi Wang [1]   Letian Li [1 2]   Guangkai Xu [1]   Donglin Huang [1]   Hao Chen [1]   Qiang Dai [2]
Weiwei Xu [1]

## Abstract

Large Language Models(LLMs) have revolutionized text generation and multimodal perception, but their capabilities in 3D content generation remain underexplored. Existing methods compromise by producing either low-resolution meshes or coarse structural proxies, failing to capture fine-grained geometry natively. In this paper, we propose CG-MLLM, a novel Multi-modal Large Language Model (MLLM) capable of 3D captioning and high-resolution 3D generation in a single framework. Leveraging the Mixture-of-Transformer architecture, CG-MLLM decouples disparate modeling needs, where the Token-level Autoregressive (TokenAR) Transformer handles token-level content, and the Block-level Autoregressive (BlockAR) Transformer handles block-level content. By integrating a pre-trained vision-language backbone with a specialized 3D VAE latent space, CG-MLLM facilitates long-context interactions between standard tokens and spatial blocks within a single integrated architecture. Experimental results show that CG-MLLM significantly outperforms existing MLLMs in generating high-fidelity 3D objects, effectively bringing high-resolution 3D content creation into the mainstream LLM paradigm. Beyond generation, we further observe that learning to produce 3D content transfers back to perception, strengthening the model's image-based 3D understanding.

## 1. Introduction

Large language models (LLMs) have achieved remarkable success through training on massive amounts of text (Team et al., 2025a; Grattafiori et al., 2024; DeepSeek-AI et al., 2025; OpenAI et al., 2024; Qwen et al., 2025; Yang et al., 2025a), offering a glimpse of hope for realizing artificial general intelligence (AGI). However, the scaling of text-only models faces a potential bottleneck as high-quality textual data nears exhaustion. Furthermore, numerous real-world dimensions are hard to capture through text alone. In response, researchers have begun extending large language models to understand and generate content across multiple modalities, achieving impressive results.

Recently, a series of Multimodal large Language Models (MLLMs) (Zhou et al., 2024; Chen et al., 2025a; Team, 2025; Comanici et al., 2025; Cui et al., 2025; Xie et al., 2025b; Deng et al., 2025) have demonstrated impressive spatial intelligence. They have revolutionized multimodal understanding and text-to-image synthesis by effectively bridging visual inputs with textual descriptions, enabling implicit spatial intelligence in 2D vision. Nevertheless, such spatial intelligence remains fundamentally constrained to 2D image space. These models do not explicitly reason about geometry, topology, or spatial consistency in 3D space, which are essential for real-world structure generation.

This limitation highlights a clear disparity between the rapid advances in 2D multimodal generation and the slow progress in 3D modeling. While progress in areas such as images, audio, and video has been booming, advancements in the 3D domain remain relatively scarce and limited, and are increasingly lagging behind those in other modalities.

To narrow this gap, prior works explored Multimodal large language models(MLLMs) for 3D generation. Existing approaches mainly follow two paradigms. The first generates meshes in textual or tokenized form (Fang et al.; Wang et al., 2024b; Dai et al., 2025), but token budget limits mesh complexity and resolution. The second constructs coarse 3D structures using low-resolution voxel VAEs or lego-based structures (Ye et al., 2025; Pun et al., 2025), producing only low-detail proxy shapes and still relying on additional 3D diffusion for fine-grained geometry. In other words, current 3D large language models are only capable of generating coarse 3D representations at the language modeling stage, and are unable to end-to-end generate detailed 3D objects.

[1]Zhejiang University, China [2]LIGHTSPEED. Correspondence to: Weiwei Xu <xww@cad.zju.edu.cn>.

*Proceedings of the 43rd International Conference on Machine Learning*, Seoul, South Korea. PMLR 306, 2026. Copyright 2026 by the author(s).

To address these challenges, we introduce CG-MLLM, a language–image–3D multimodal large language model, with the goal of using a single model to perform precise spatial understanding and generate high-fidelity spatial content with strong 3D consistency. Compared with existing 3D generation methods, which either do not incorporate language modeling at all, or cannot natively produce high-resolution 3D content within the LLM framework, CG-MLLM emphasizes native language-image-3D integration. In this way, we aim to bring 3D creation closer to the mainstream MLLM paradigm, enabling the 3D domain to more directly benefit from the rapid progress and scaling momentum of LLMs.

The core challenge in 3D MLLMs is effectively modeling 3D geometry, which naturally forms long, highly interdependent sequences. A purely token-level autoregressive formulation inevitably leads to severe inefficiency. Inspired by Mixture-of-Transformers(MoT), we adapt dedicated transformers for Token-level serial modeling (TokenAR) and Block-level parallel modeling (BlockAR). Unlike existing MoT-based methods that rigidly bind transformers to specific tasks (e.g., understanding or generation), our CG-MLLM architecture is fundamentally different in that it binds them to their generation modes (e.g., serial or parallel). This design allows any encoder to be plugged into its corresponding transformer according to its native pre-training scheme, while keeping the transformer's input and masking operations—akin to LoRA zero initialization—minimally perturbing the pre-trained model and reducing fine-tuning cost. By integrating a pretrained Qwen3-VL (Bai et al., 2025) backbone with a high-order latent space provided by Hunyuan3D2.1-VAE (Hunyuan3D et al., 2025), CG-MLLM enables the simultaneous processing of sequential linguistic intent and parallel block-level 3D construction. This architecture not only preserves global spatial consistency but also significantly enhances the efficiency of end-to-end 3D generation.

In summary, CG-MLLM demonstrates a clear superiority over the state-of-the-art methods in understanding tasks and achieves the best generation quality among existing LLM-based 3D models. Moreover, equipping the model with 3D generation not only enables high-fidelity 3D synthesis, but also yields a concurrent improvement in its ability to interpret 3D structure from 2D images.Our main contributions are as follows:

- We propose a new end-to-end 3D generative method natively integrated with a Large Vision-Language Model.
- We propose CG-MLLM, which outperforms Qwen3-VL in 3D understanding tasks, demonstrating superior spatial reasoning, while also achieving state-of-the-art performance in LLM-based 3D generation.
- We bridge multimodal LLMs with high-precision 3D synthesis at a lower computational cost, our approach

provides a highly efficient and scalable architecture for future research in generative 3D modeling.

## 2. Related Work

### 2.1. Autoregressive Models

Large Language Models (LLMs) (Team et al., 2025a; Grattafiori et al., 2024; DeepSeek-AI et al., 2025; OpenAI et al., 2024; Qwen et al., 2025; Yang et al., 2025a) represent the most prominent and widely adopted class of autoregressive (AR) generative models. By sequentially predicting each element conditioned on preceding ones, AR models are highly effective across a broad spectrum of real-world applications. Recent advancements in Large Language Models (LLMs) have catalyzed the development of Multimodal Large Language Models (MLLMs), which demonstrate exceptional performance across diverse scenarios. A standard implementation involves projecting the outputs of multimodal encoders into the LLM's embedding space, effectively aligning textual and multimodal representations, allowing the model to generate captions for images, video, and other multimodal content. Most multimodal understanding models utilize autoregressive (AR) generation, enabling the model to perceive and interpret complex visual and linguistic information (Bai et al., 2025; Xu et al., 2025; Lu et al., 2024). Recent research has validated the effectiveness of AR frameworks in tasks such as vision and audio understanding, proving their robustness and versatility in processing cross-modal data. Furthermore, certain advanced models are capable of generating discrete multimodal tokens—exemplified by Emu3 for images (Wang et al., 2024a) and SAR3D for 3D assets (Chen et al., 2025b)—which are subsequently reconstructed into high-fidelity content via specialized decoders (Tian et al., 2024; Yin et al., 2023; Siddiqui et al., 2023). By leveraging the AR mechanism and pretrained LLM backbones, these models exhibit remarkable versatility in tasks like image captioning, video understanding, and multimodal generation; however, empirical evidence suggests that the synthesis quality of AR models still lags behind that of diffusion-based models (Deng et al., 2025). Additionally, their inherent sequential nature results in significant inference latency, posing challenges for real-time applications.

### 2.2. Diffusion Models

Unlike autoregressive (AR) models, which operate on discrete tokens, diffusion models work directly with continuous vectors, demonstrating remarkable generative capabilities through a combination of forward noise addition and learned reverse denoising. This property makes diffusion-based approaches particularly well-suited for multimodal generation tasks, including image (Liu et al., 2022; Lipman et al., 2023; Esser et al., 2024; Peebles & Xie, 2023; Wu et al., 2025b),

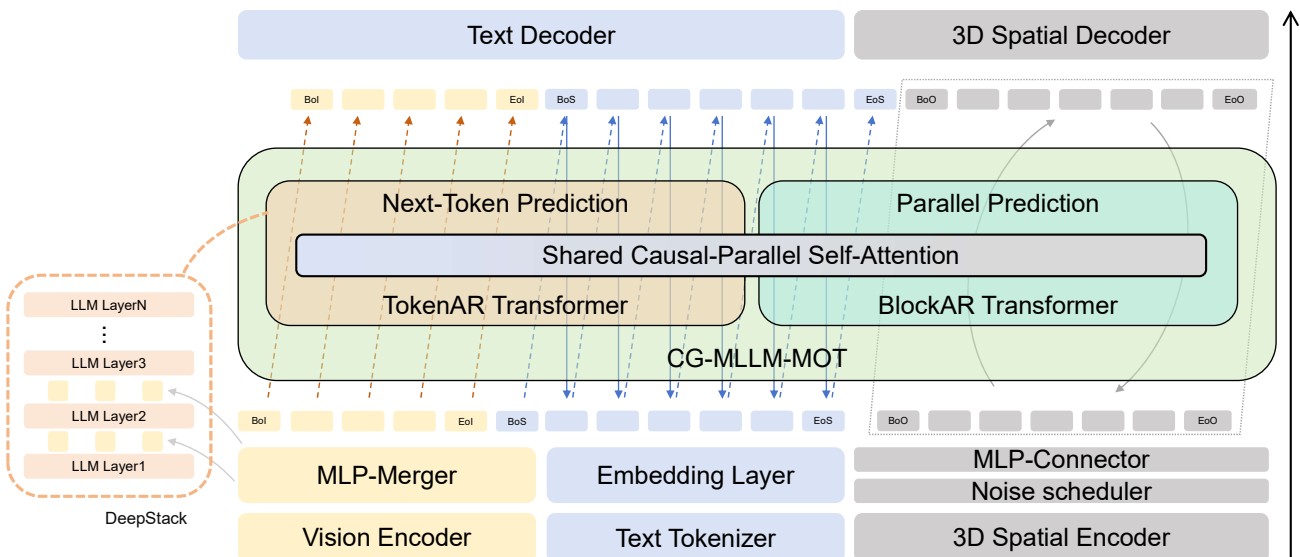

*Figure 1.* The Pipeline of CG-MLLM. Our multimodal architecture processes vision, text, and 3D spatial inputs to generate text and 3D spatial outputs. It features a TokenAR Transformer for sequential next-token prediction and a BlockAR Transformer for efficient parallel block prediction, both governed by strict causal masking.

video (Wan et al., 2025; Wu et al., 2025a; Hong et al., 2022; Yang et al., 2025b; Ma et al., 2025a), and 3D content (Cheng et al., 2023; Zhao et al.; Zhang et al., 2024; Zhao et al., 2025; Xiang et al., 2025; Hunyuan3D et al., 2025; Li et al., 2025b;a;c; Feng et al., 2025) synthesis.

The diffusion paradigm consists of two key stages: a forward process that gradually transforms data into noise, and a reverse process, learned by a neural network, which reconstructs data from noise. To mitigate the complexity associated with traditional diffusion schemes, recent works have increasingly adopted the Rectified Flow framework (Liu et al., 2022; Lipman et al., 2023; Esser et al., 2024). The workflow of this framework can be summarized as follows:

**Probability Path Construction.** Given a data sample $x_0 \sim P_{data}$ and a noise sample $\epsilon \sim N(0, I)$, a linear interpolation defines the intermediate state $x_t$ at time $t \in [0, 1]$:

$$x_t = (1 - t)x_0 + t\epsilon$$

**Velocity Field Modeling.** Rectified Flow targets the constant velocity of the linear trajectory:

$$v = \frac{dx_t}{dt} = \epsilon - x_0$$

A neural network $v_\theta(x_t, t)$ is parameterized to approximate this velocity field via a Mean Squared Error (MSE) objective:

$$\mathcal{L} = \mathbb{E}_{x_0, \epsilon, t} \left[ \|v - v_\theta(x_t, t)\|^2 \right]$$

**Inference.** During the inference stage, starting from pure noise $\epsilon$ ($t = 1$), the data $x_0$ ($t = 0$) is recovered by nu-

merically solving the reverse Ordinary Differential Equation (ODE):

$$\frac{dx_t}{dt} = v_\theta(x_t, t)$$

This formulation significantly enhances sampling efficiency and reduces the truncation errors typically associated with curved diffusion paths.

### 2.3. AR-Diffusion Models

Recent research has extensively explored the integration of AR models and Diffusion models within a single framework, aiming to leverage the strengths of AR models in discrete sequence modeling alongside the superior continuous distribution synthesis of Diffusion models. Such hybrid architectures inherit the profound semantic priors of LLMs for understanding, while maintaining the iterative refinement capabilities of diffusion processes for high-fidelity generation. Generally, these hybrid methodologies can be categorized into two distinct paradigms:

**Decoupled AR-Diffusion Pipelines** This approach treats the LLM as a high-level semantic controller that interfaces with an external diffusion-based generator (Wu et al., 2025b; Ye et al., 2025; Dong et al., 2024; Ge et al., 2025). The LLM interprets user intent and generates latent semantic conditions, which are fed into a pretrained diffusion module for content synthesis. While this paradigm benefits from lower computational overhead and the ability to leverage existing pretrained experts, the inherent decoupling of understanding and generation may lead to information bottlenecks and a lack of fine-grained cross-modal alignment.

**Integrated AR-Diffusion Pipelines** A more integrated paradigm involves modifying the LLM's internal architecture, specifically by transitioning from causal masking to bidirectional or parallel masking, to enable native diffusion capabilities within the same backbone (Zhou et al., 2024; Deng et al., 2025; Ma et al., 2025b; Xie et al., 2025a;b; Cui et al., 2025). Unlike the decoupled approach, this strategy maintains a lossless context across all modalities, fostering seamless interaction between understanding and generation. Although it demands significantly higher computational resources for training, its scalability and architectural consistency offer a more robust path toward general-purpose multimodal intelligence.

## 3. Method

As illustrated in Fig. 1, the architecture of CG-MLLM deploys a decoder-only transformer architecture and follows a MoT (Liang et al., 2025) design, built upon a powerful pre-trained VLM backbone. The architecture primarily consists of three functional stages: multimodal encoding, specialized MoT modeling, and multimodal decoding. In the encoding stage, modality-specific encoders transform heterogeneous inputs—including text, images, and 3D assets—into a unified token space. These tokens are then processed in the MoT modeling stage, where a dual-transformer system (TokenAR and BlockAR) performs joint multimodal reasoning and spatial sequence generation via a shared attention mechanism. Finally, the latent representations are passed to the decoding stage: a text head decodes linguistic tokens into natural language, while a dedicated 3D decoder maps the spatial block tokens back into high-fidelity 3D representations. This end-to-end design enables CG-MLLM to achieve seamless perception and generation across both textual and 3D spatial modalities.

### 3.1. Modality Adapters

To facilitate effective multimodal integration, we employ distinct tokenization strategies for different modalities, as detailed below:

**Text Tokenization.** For textual input, we utilize Qwen's tokenizer consistent with the baseline VLM, which implements byte-level Byte-Pair Encoding (BBPE (Yang et al., 2025a; Wang et al., 2019)) with a vocabulary size of 151,669.

**Visual Tokenization.** Following the architecture of Qwen3-VL (Bai et al., 2025), we leverage a SigLIP-2 (Tschannen et al., 2025) encoder for image feature extraction. To accommodate various input resolutions, we adopt its strategy of employing 2D-RoPE (Su et al., 2023) and interpolating absolute position embeddings. Furthermore, a two-layer MLP is utilized to compress $2 \times 2$ visual features into a single visual token. This design aligns the visual tokens with

the LLM's hidden dimension and supports the DeepStack mechanism for efficient multi-scale processing.

**3D Tokenization.** To enable the perception and generation of 3D content, we integrate a pre-trained Spatial-VAE adapted from Hunyuan3D-2.1 (Hunyuan3D et al., 2025). This component extracts point clouds from 3D objects surfaces and encodes them into a high-dimensional latent space via an attention-based mechanism. The Spatial-VAE operates with a downsampling factor of 20 and a latent dimension of 64. These latent representations are subsequently aligned with the multimodal semantic space through a dedicated Connector layer before being fed into the LLM. To maintain the stability of the learned geometric priors, the Spatial-VAE remains frozen during the training process.

### 3.2. Backbone

Specifically, our backbone is built upon the Qwen3-VL architecture, benefiting from its superior pretrained performance and robust ecosystem. Both the TokenAR and BlockAR transformers are initialized with pre-trained Qwen3-VL weights, allowing CG-MLLM to leverage rich multimodal priors while facilitating functional specialization. While TokenAR maintains the model's original capacity for token-level autoregressive modeling, BlockAR extends this design to handle parallel block tokens. This dual-transformer design enables CG-MLLM to inherit the extensive knowledge of the VLM for 2D visual understanding while simultaneously empowering it with robust spatial perception and generative capabilities within a unified framework. Moreover, BlockAR enables efficient parallel processing of large numbers of tokens. Our experiments show that when targeting a spatial latent resolution of 4096 tokens, the block-level approach achieves a threefold speedup compared with token-level processing.

**Hybrid Receptive Field.** To synergize sequential token generation with parallel block modeling, we employ a hybrid masking mechanism that transcends the limitations of traditional singular masking strategies. Unlike standard autoregressive models that rely solely on causal masks (where tokens attend only to preceding ones) or diffusion-based models that utilize parallel masks (where tokens attend to all tokens within a sample), CG-MLLM adaptively combines both. To further illustrate, the specific masking patterns applied across different modalities and tasks are visualized in Fig. 3. In this heatmap, each colored cell in a given row represents the set of previous or concurrent tokens that the corresponding token is permitted to attend to, effectively defining its attention visibility within the hybrid framework.

**Architectural Components.** These transformers inherit the core components of Qwen3-VL (Bai et al., 2025), including Grouped Query Attention (GQA) (Ainslie et al., 2023), SwiGLU activation (Dauphin et al., 2017), and RM-

Figure 2. Our approach unifies spatial perception and generation in a single model, supporting image understanding, point cloud understanding, mesh generation, and textual intent understanding across multiple spatial modalities.

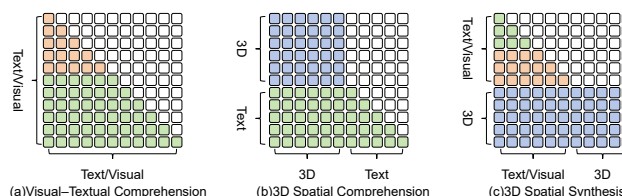

Figure 3. Example mask used in CG-MLLM.

SNorm (Zhang & Sennrich, 2019). To improve multimodal alignment stability, we leverage Interleaved Multimodal Rotary Positional Embeddings (Interleaved MRoPE) (Bai et al., 2025; Su et al., 2023) and QK-Norm (Dehghani et al., 2023) from the Qwen3-VL (Bai et al., 2025) framework.

### 3.3. Modality Decode

After generation, the predicted tokens are decoded into their respective modalities. Text tokens are directly decoded using the tokenizer described in Sec. 3.1. For 3D outputs, we decode the tokens into geometric representations via the VAE of Hunyuan3D-2.1 (Hunyuan3D et al., 2025), followed by material synthesis using its corresponding material generator for improved visual fidelity.

### 3.4. Comparison with Related MoT Methods

Our architecture is conceptually inspired by pioneering methods (Liang et al., 2025; Shi et al., 2025; Deng et al., 2025), yet it represents a significant evolution in several fundamental aspects: (a) Transformer Specialization Logic: Unlike existing methods (Shi et al., 2025; Deng et al., 2025), which partition transformers based on functional splits (i.e., understanding vs. generation), CG-MLLM adopts a model-based binding strategy. Under this scheme, TokenAR is bound to sequential token-level modeling, while BlockAR

is bound to parallel block-level synthesis. This logic fundamentally differs from task-binding paradigms, enabling flexible integration of encoders according to their original pre-training schemes. The design also maintains extensibility, readily supporting potential future extensions such as autoregressive latent generation and diffusion-based text understanding. (b) Vision-Centric Initialization: Unlike existing methods (Shi et al., 2025; Deng et al., 2025), which require costly retraining of visual comprehension for the language-centric methods, CG-MLLM leverages the vision-centric Qwen3-VL (Bai et al., 2025). By inheriting the advanced DeepStack (Meng et al., 2024) mechanism, our framework captures finer visual details and achieves better alignment (Bai et al., 2025). This strategic inheritance allows CG-MLLM to capitalize on mature VLM capabilities while drastically reducing training overhead, providing a robust foundation for geometric reasoning. (c) Positional Encoding Strategy: We introduce a 3D-aware strategy. Beyond inheriting Interleaved MRoPE and 2D-RoPE from Qwen3-VL (Bai et al., 2025), we intentionally omit intra-block positional embeddings for 3D tokens. By assigning a distinct block-level position while sharing an identical positional index within each block, we preserve the permutation invariance of point features while maintaining global spatial structure. (d) Volumetric Spatial Intelligence: While existing methods (Liang et al., 2025; Shi et al., 2025; Deng et al., 2025) are primarily restricted to 2D visual-text representations, CG-MLLM evolves towards true spatial intelligence. By integrating 3D geometric priors into the VLM, our framework transcends flat pixel-space, enabling the model to perceive, reason about, and synthesize complex 3D structures with physical consistency.

## 3.5. Training Recipe

Our training consists of two stages. In the first stage, the alignment stage, after initializing the model, we first train its unconditional generation capability along with the initial understanding ability. Specifically, we discard 90% of the conditional inputs and train at a 3D resolution of 512 tokens.

In the second stage, the progressive resolution stage, we gradually increase the resolution from 512 to 4096 tokens, while reducing the discard probability from 90% to 10%.

## 4. Experiments

### 4.1. Dataset

We curated our dataset from LLaVA-OneVision (Li et al.), Trellis-500K (Xiang et al., 2025), and Objaverse++ (Deitke et al., 2023; Lin et al., 2025).Following the protocol of prior work(Xiang et al., 2025), we adopt the toys4k subset of Trellis-500K as our held-out test set. For the Objaverse++ (Lin et al., 2025) subset, we selected samples with the highest aesthetic rating (score=3) and used their corresponding rendered images from Objaverse-MIX (Qian et al., 2024). For the Trellis-500K dataset, we utilized its rendering pipeline and adopting the Hunyuan3D2.1 pipeline for watertight processing and surface sampling. We initially followed the Hunyuan3D2.1 configuration, pre-sampling 124,928 for both uniform points and importance points and randomly selecting from these points during data loading, but ultimately used only the uniform points, as explained in Section 4.4.

### 4.2. Implementation Details

We employ Classifier-Free Guidance (CFG) with a dropout rate that varies during training. The model follows a progressive training strategy, in which the sequence length is gradually increased from 512 to 4096 tokens, as mentioned in Section 3.5. We use the AdamW optimizer, adjusting the learning rate from $1 \times 10^{-4}$ down to $5 \times 10^{-5}$ as the token count increases. For the diffusion process, we apply the logit-normal sampler (Esser et al., 2024) for the timesteps, maintaining a scale of 1.0 throughout the training duration. The training was conducted on 16 NVIDIA H20 GPUs, with the maximum sequence length increasing from 36,864 to 51,200 as the 3D token resolution grows.During inference, we set the CFG scale to 7.5 and perform 50 sampling steps.

### 4.3. Limitations of AdaLN in MLLM

Given the outstanding performance of Adaptive Layer Normalization(AdaLN) (Peebles & Xie, 2023) in various diffusion tasks and the capability of MoT to decouple text from multimodal content, we initially explored integrating AdaLN into the multimodal branches. Specifically, we ap-

plied the $(1 + scale) \times hidden + shift$ transformation following the RMSNorm layer of the LLM, rather than the standard LayerNorm. To ensure that the initial training steps remained unaffected, the linear layers responsible for generating scale and shift parameters were zero-initialized. Unfortunately, as illustrated in Fig. 6(a), the training loss with AdaLN was substantially higher than the baseline. This suggests that incorporating AdaLN into multimodal large language models (MLLMs) is suboptimal in this context. We hypothesize that the introduction of additional scaling factors may compromise the stability of the Shared Causal-Parallel Self-Attention mechanism.

### 4.4. VAE Reconstruction Issues

We observe that when using the Hunyuan3D-2.1 VAE (Hunyuan3D et al., 2025), following the point sampling strategy proposed in their paper, i.e., a mixture of uniformly sampled points and importance points, can lead to distorted reconstructions with holes in some cases. This issue is common, especially when only a small number of points are randomly sampled from pre-sampled Farthest Point Sampling(FPS) points, as illustrated in Fig. 6(b). The examples on the left show the reconstruction results without importance points, while those on the right use importance points.

Therefore, we only use uniformly points in our experiments to ensure stable and reliable reconstructions.

### 4.5. Quantitative comparison

We evaluate our generated results using p-FID and p-KID (Heusel et al., 2018; Bińkowski et al., 2021; Nichol et al., 2022) to compare the distribution similarity between the generated geometries and the ground truth. We adopt CLIP-IQA+ (Wang et al., 2022) and MUSIQ (Ke et al., 2021) to assess the visual quality of the final outputs, and CLIP (Radford et al., 2021) to measure the consistency between the outputs and the inputs. In addition, we collected 100 questionnaires, resulting in approximately 2,400 data points for a statistically meaningful analysis of user preferences. Compared with earlier MLLM-based 3D generation methods, such as SAR3D (Chen et al., 2025b) and ShapeLLM-Omni (Ye et al., 2025), our approach consistently outperforms them across all quantitative metrics, as shown in Tab. 1, achieving the best performance among compared MLLM-based methods.Nevertheless, we acknowledge that although our method represents a state-of-the-art MLLM-based 3D generation model (Xiang et al., 2025; Zhao et al.; Li et al., 2025a; Hunyuan3D et al., 2025; Team et al., 2025b), it still does not comprehensively surpass state-of-the-art non-MLLM-based methods. Closing this gap and ultimately outperforming such methods remains an open problem, with promising directions including stronger multi-encoder image conditioning, scaling up model param-

*Table 1.* Performance comparison with Non-MLLM-based and MLLM-based 3D generative models. The best results in each column are highlighted in red, while the second-best results are highlighted in yellow.

| Method | p-FID↓ | p-KID↓ | CLIP-IQA+↑ | MUSIQ↑ | CLIP↑ | User Study↑ |
|---|---|---|---|---|---|---|
| *Non-MLLM-Based* | | | | | | |
| Michelangelo | 17.96 | 0.56 | 0.45 | 71.42 | 84.08 | 2.60 |
| CraftsMan | 14.09 | 0.40 | 0.45 | 71.09 | 84.86 | 3.15 |
| Hunyuan3D-2.1 | 16.80 | 0.53 | 0.47 | 71.20 | 85.11 | 3.15 |
| TRELLIS | 7.36 | 0.12 | 0.44 | 66.97 | 84.13 | 3.28 |
| SAM3D | 33.92 | 1.13 | 0.47 | 70.21 | 84.67 | 3.45 |
| *MLLM-Based* | | | | | | |
| SAR3D | 30.07 | 1.00 | 0.42 | 66.01 | 82.86 | 2.93 |
| ShapeLLM-Omni | 13.11 | 0.29 | 0.37 | 55.71 | 84.18 | 2.30 |
| Ours | 12.55 | 0.27 | 0.45 | 71.65 | 84.47 | 3.32 |

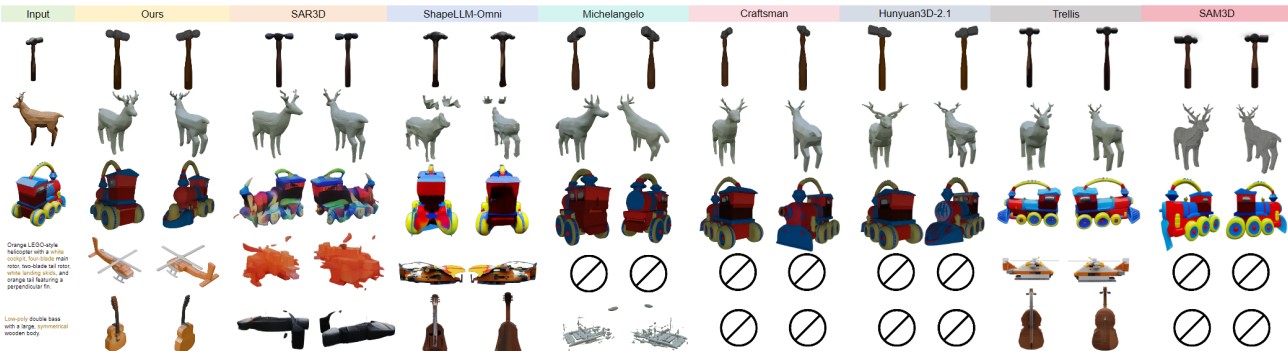

*Figure 4.* Comparison with other methods on the image-to-3D task. For clearer visualization of geometry, materials are removed in the second row. Compared with MLLM-based methods, our method generates more complete geometry, while achieving comparable visual quality to non-MLLM methods.

eters, and increasing the 3D token budget.

Although our design supports 3D understanding, most captions in the dataset we use (Xiang et al., 2025) (usually less than 20 words) are generated by an MLLM, limiting the model's 3D understanding, as illustrated in Fig. 2. Therefore, when designing the spatial understanding functionality, we retain the image-based understanding component and treat it as the primary modality for interpreting rendered 3D geometry. Quantitative results for spatial content captioning are reported in Tab. 2. Compared to LLaVA (Liu et al., 2023), InstructBLIP (Dai et al., 2023), and the Qwen3-VL(Bai et al., 2025), our method achieves consistently better performance across all metrics. This indicates that jointly training for 3D generation not only equips the model with generative capability, but also transfers back to perception, enhancing its ability to reason about 3D structure from images alone. However, our method currently trails established approaches such as 3D-LLM (Hong et al., 2023), PointLLM (Xu et al., 2024), and ShapeLLM-Omni-7B (Ye et al., 2025). This performance gap is likely attributable to their extensive 3D captioning datasets and larger model scale.

| Model | BLEU-1↑ | ROUGE-L↑ | METEOR↑ |
|---|---|---|---|
| *3D latent Inputs* | | | |
| 3D-LLM | 16.91 | 19.48 | 19.73 |
| PointLLM-13B | 17.09 | 20.99 | 16.45 |
| ShapeLLM-Omni-7B | 18.51 | 21.37 | 19.89 |
| *Image Inputs* | | | |
| InstructBLIP-13B | 4.65 | 8.85 | 13.23 |
| LLaVA-13B | 4.02 | 8.15 | 12.58 |
| Qwen3-VL-2B | 3.13 | 7.21 | 11.92 |
| Ours-2B MOT | 13.51 | 19.13 | 14.28 |

*Table 2.* Quantitative comparison on 3D object captioning metrics. The best results in each column is highlighted in red, while the second-best results is highlighted in yellow.

### 4.6. Qualitative comparisons

Our method demonstrates strong performance in both geometric fidelity and semantic alignment. In comparisons with existing approaches such as SAR3D (Chen et al., 2025b) and ShapeLLM-Omni (Ye et al., 2025), our results exhibit superior geometric completeness and conditional consistency as shown in Fig. 4. Further examples, provided in Fig. 5 and the appendix, show the method consistently generates detailed, coherent 3D geometry. Moreover, our captioning

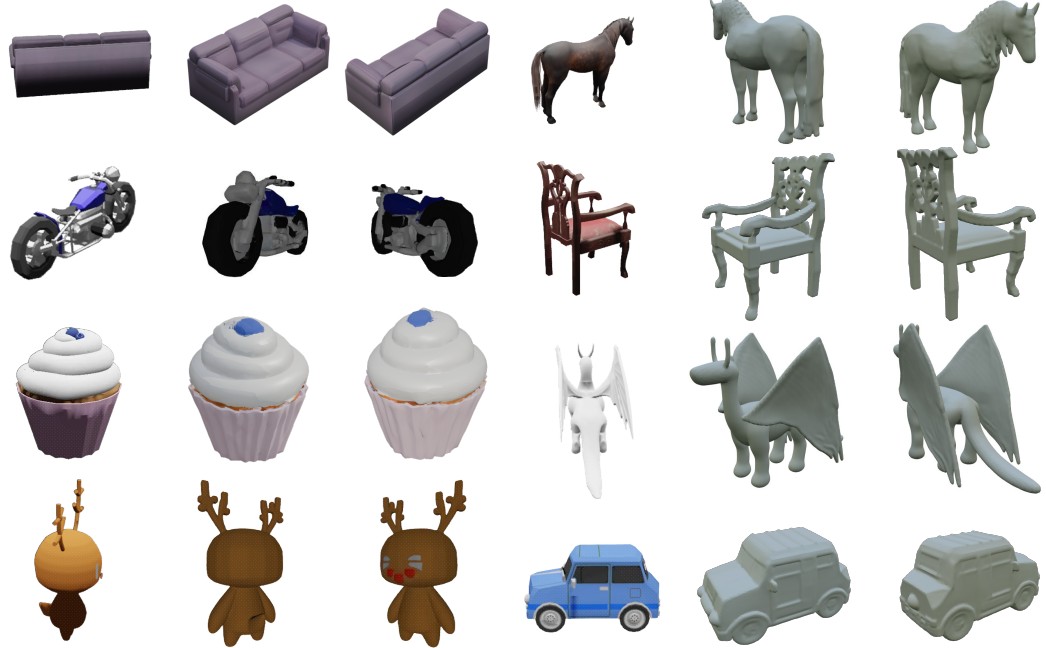

*Figure 5.* More Image-to-3D results produced by our method. For clearer visualization of geometry, materials are removed from the rightmost column. Zooming in is recommended for better inspection.

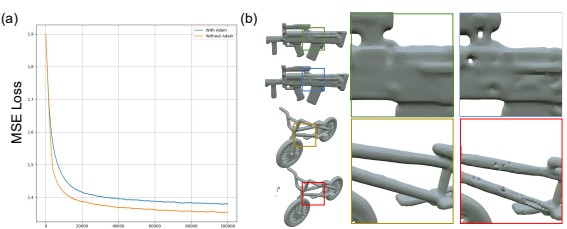

*Figure 6.* Comparison of different strategies. (a) Training MSE loss comparison w/ and w/o AdaLN under the LLM architecture. (b) VAE reconstruction comparison w/ and w/o importance points.

*Table 3.* Ablation studies on CG-MLLM.

| HY2.1-VAE | MOT | LLM Backbone | #Tokens | p-FID↓ | p-KID↓ |
|:---:|:---:|:---|:---:|:---:|:---:|
| ✗ | ✗ | Qwen2.5-0.5B | 512 | 53.66 | 1.76 |
| ✓ | ✗ | Qwen2.5-0.5B | 512 | 44.91 | 1.42 |
| ✓ | ✓ | Qwen2.5-0.5B | 512 | 30.60 | 0.77 |
| ✓ | ✓ | Qwen3VL-2B | 512 | 15.61 | 0.43 |
| ✓ | ✓ | Qwen2.5-0.5B | 4096 | 16.57 | 0.53 |
| ✓ | ✓ | Qwen3VL-2B | 4096 | **12.55** | **0.27** |

results (Fig. 7) outperform the ground-truth captions used for 3D captioning evaluation in several cases, demonstrating the capability of our approach to produce accurate and fine-grained textual descriptions.

### 4.7. Ablation studies

We conduct five groups of ablation studies on the key design choices of our method: (1) VAE backbone: comparing Hunyuan3D-2mini against Hunyuan3D-2.1 to isolate the impact of VAE representation capacity on the overall performance; (2) MOT architecture: verifying the necessity of MOT, while the specific design choice within MOT (standard DiT vs. LLM-style BlockAR) has already been analyzed in Sec. 4.3; (3) LLM backbone: replacing Qwen3-VL with Qwen2.5-0.5B to evaluate the portability of our method to smaller backbones; (4) OBJ token length: examining how different 3D-token lengths affect generation

quality under the same model size; and (5) Block-size / token-budget scaling: providing speed and memory curves over the range from 512 to 5120 tokens. As shown in Tab. 3, the stronger Hunyuan3D-2.1 VAE, the MOT architecture, and a larger 3D-token budget all yield consistent gains; replacing Qwen2.5-0.5B with Qwen3-VL 2B further improves performance, indicating that our method follows the scaling-law trend. Tab. 4 reports the training cost under a fixed budget of 20,480 tokens per GPU on 8 GPUs: as the number of 3D tokens increases, the total number of samples per batch (including those processed by the multi-modal encoders) decreases, and the memory consumption drops accordingly. Tab. 5 summarizes how the inference cost scales with 3D-token length: memory grows linearly, while runtime scales approximately quadratically with a small leading constant.

*Table 4.* Training cost on 8 GPUs with a fixed budget of 20,480 tokens per GPU. "#Samples" is the total number of samples per batch; "#3D" counts only the 3D-generation samples.

| #3D Tokens | Step/s | Memory (GB) | #Samples | #3D |
|---|---|---|---|---|
| 512 | 0.26 | 75 | 300 | 118 |
| 1024 | 0.27 | 60 | 230 | 78 |
| 2048 | 0.27 | 54 | 150 | 50 |
| 4096 | 0.22 | 43 | 100 | 32 |
| 5120 | 0.20 | 40 | 90 | 24 |

*Table 5.* Inference cost as a function of 3D-token length.

| #3D Tokens | 512 | 1024 | 2048 | 4096 | 5120 |
|---|---|---|---|---|---|
| Time (s) | 12 | 13 | 19 | 35 | 45 |
| Memory (GB) | 9 | 9 | 10 | 11 | 12 |

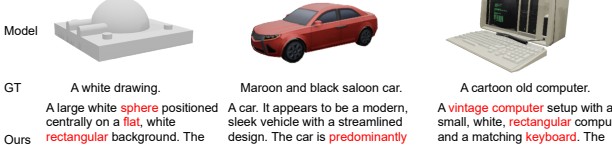

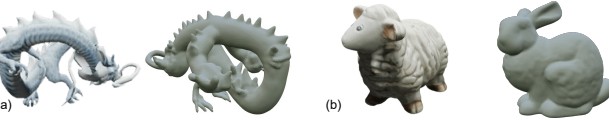

*Figure 7.* Caption results. Compared to the ground truth from the point cloud perception dataset our descriptions are more detailed and showcase the model's spatial reasoning capabilities.

*Figure 8.* Failure Cases. (a) Common: ambiguous hints often lead to inaccurate outputs. (b) Rare: input and output are semantically similar but differ greatly in details.

## 5. Limitation

When the input is ambiguous or potentially confusing, our model may produce suboptimal results, as shown in Fig. 8 (a). This type of failure is relatively common, and we observed artifacts even when processing this case with multiple commercial 3D generation models. In addition, we identified a rare failure case on our model, illustrated in Fig. 8 (b), where the hint image depicts a sheep, but the generated 3D model is a rabbit. Since our vision encoder operates at the semantic level rather than the pixel or feature level, and considering that MLLMs occasionally misclassify semantics, we attribute this rare failure to hallucination.

Beyond these specific failure cases, our method still falls short of leading commercial 3D generation systems in overall quality. This gap likely stems from our data preprocessing and 3D reconstruction pipeline based on Hunyuan3D 2.1 (Hunyuan3D et al., 2025), which first thickens the model and converts it into a watertight mesh. While this step ensures topological correctness, it inevitably reduces data precision. In addition, the reconstruction uses relatively few tokens (fewer than 4k) whereas current high-resolution 3D generation methods often require up to ten times more tokens, posing challenges for LLM-based generation. We anticipate that the development of more lightweight and efficient 3D-VAEs will enable substantial improvements in our model's capabilities.

## 6. Conclusion

In this work, we propose CG-MLLM, the first multi-modal large language model (MLLM) supporting end-to-end 3D spatial generation. By integrating Qwen3-VL with Hunyuan3D2.1-VAE, we demonstrate that an MLLM can perform naive 3D generation end-to-end, without relying on external generators. Experimental results highlight the effectiveness of CG-MLLM in both 3D captioning and 3D generation, showing its capability to understand and generate spatial content and achieving state-of-the-art performance among LLMs for spatial generation. Future work can further explore scalability, aiming toward a fully unified model for multimodal generation and understanding.

## Impact Statement

This paper presents work whose goal is to advance the field of machine learning. There are many potential societal consequences of our work, none of which we feel must be specifically highlighted here.

## Acknowledgements

We sincerely thank the anonymous reviewers for their professional, insightful, and constructive comments, which have helped us improve the quality and clarity of this paper. Weiwei Xu is partially supported by NSFC (92570206, 62421003), and the State Key Lab of CAD&CG, Zhejiang University.

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

## A. More results produced by our method.

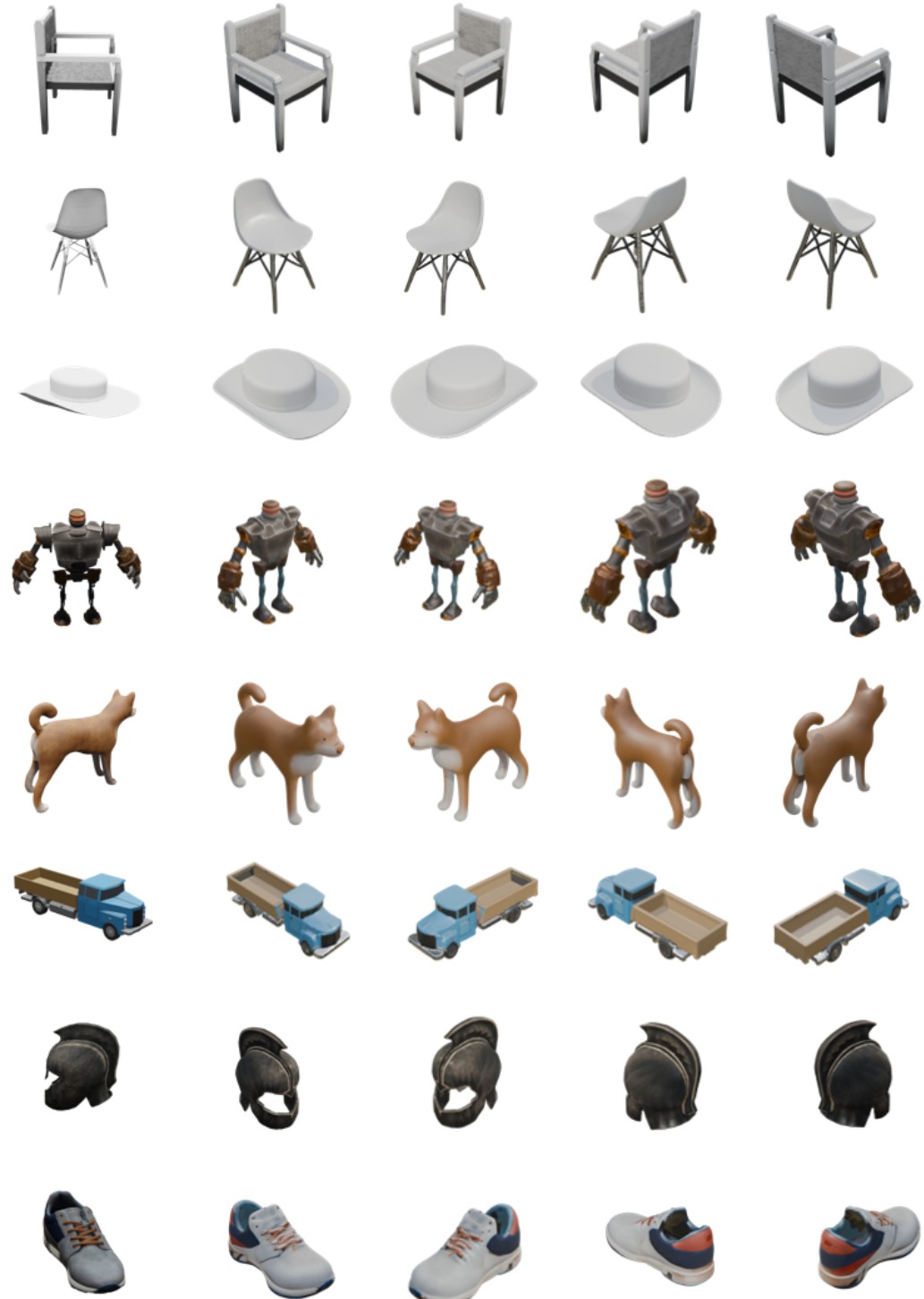

*Figure 9.* More results produced by our method.

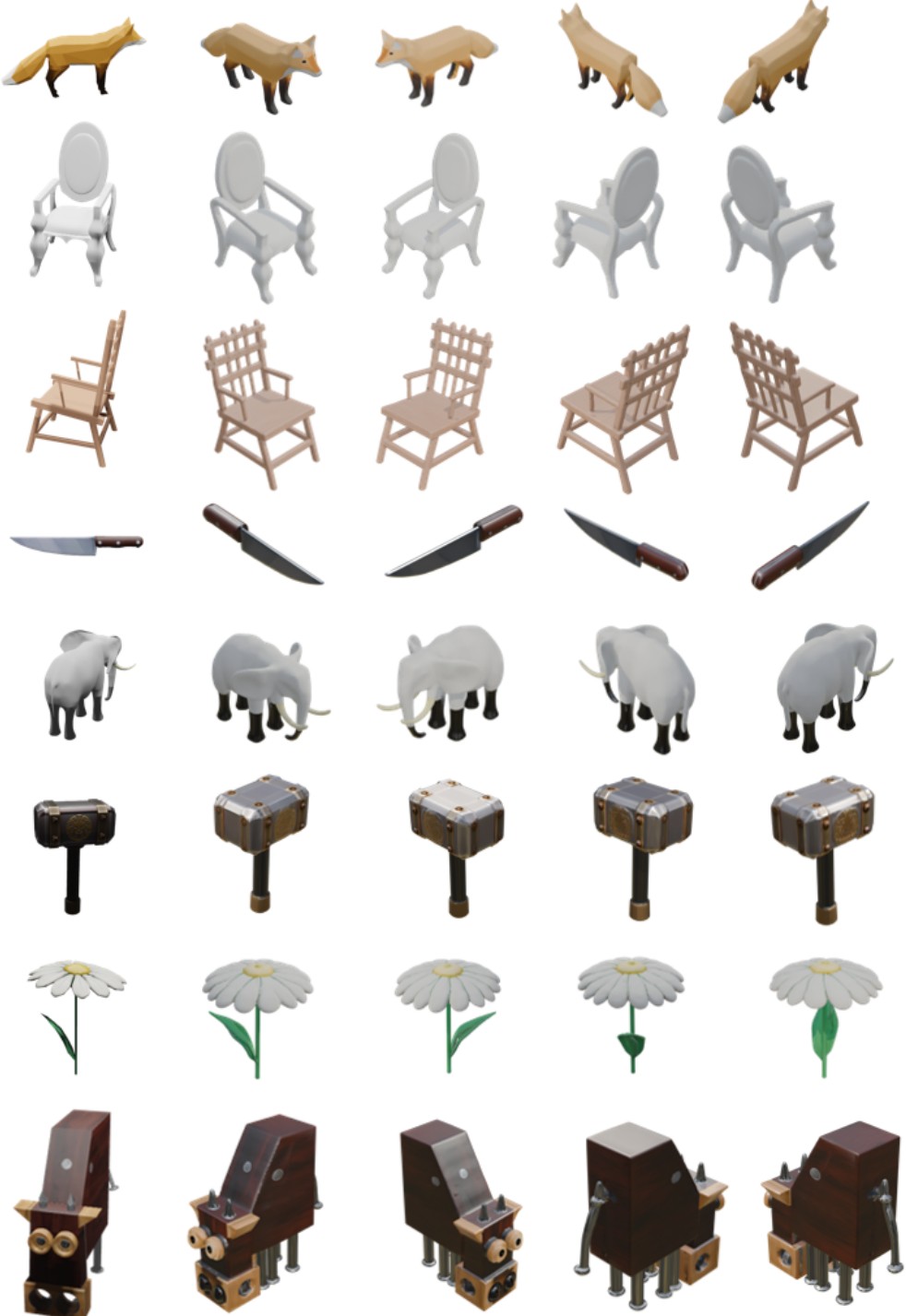

*Figure 10.* More results produced by our method.

A mid to large-sized dog with an elongated body, ... uniformly glossy black.

A dark brown horse with an elongated triangular head,...a long flexible tail.

A classic cylindrical metal trash can with a slightly domed circular lid featuring a straight rectangular handle..

Brown high-top sneaker with black accents, leather upper, padded ankle collar, ...

Wooden armchair with ornate carvings, reddish-brown polished frame....

A cylindrical, ribbed black bottle with a green screw-top cap, slightly tapered towards the top, ...

A bowler hat with a rounded dome-shaped crown and a narrow brim that curves slightly upward at the edges. ...

A nearly spherical apple with a slightly rough texture, ...,and a short, brownish stem.

*Figure 11.* More results produced by our method.

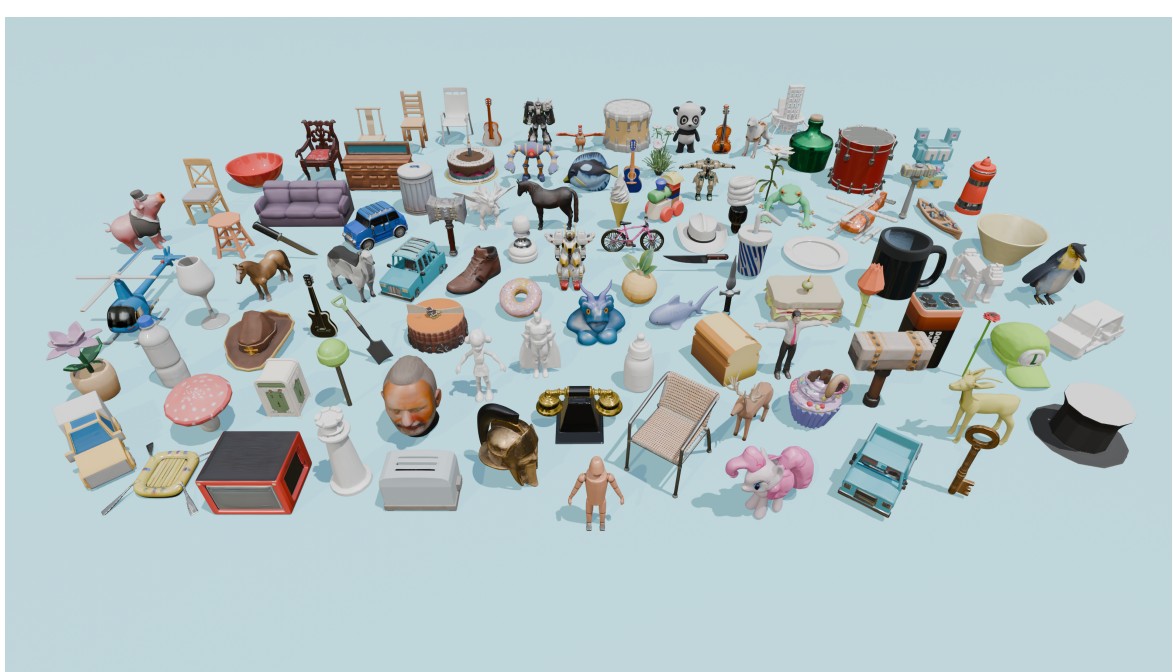

*Figure 12.* More results produced by our method.

