# OpenReview forum: "CG-MLLM: Captioning and Generating 3D content via Multi-modal Large Language Models"
_ICML.cc/2026/Conference — ICML 2026 regular_

### Official Review · Reviewer_WZkr · 2026-03-11

**Soundness:** 3
**Presentation:** 3
**Significance:** 3
**Originality:** 3
**Overall Recommendation:** 4
**Confidence:** 3

**Summary:**

CG-MLLM is a unified language–image–3D multimodal LLMj, which both understands and generates 3D content in an end-to-end way. It introduces a Mixture-of-Transformers tied to generation modes: TokenAR for sequential text tokens, and BlockAR for parallel 3D block tokens, combined via a hybrid causal and parallel attention mask. Built on Qwen3-VL with SigLIP-2, it integrates a Hunyuan3D-2.1 Spatial‑VAE so the LLM directly predicts discrete 3D latent tokens, which are then decoded into geometry and materials.

With progressive training (from 512 to 4096 3D tokens) and CFG, it outperforms present MLLM-based 3D methods on p-FID, p-KID, CLIP-IQA+, and MUSIQ, and achieves  about 3 times faster 3D generation at 4096 tokens via block-level prediction.

**Compliance With Llm Reviewing Policy:**

Affirmed.

**Final Justification:**

Thanks the authors for the rebuttal in detail. My concerns are addressed with evidences, i.e., the training/inference cost and more ablation studies. I tend to accept this submission.

**Key Questions For Authors:**

1. How does BlockAR performance scale with block size and token budget (more than 4k)? Please share curves on quality, speed, and memory to justify the chosen configuration.

2. What are the failure modes when conditioning on ambiguous or noisy images or long instructions, and do you have mechanisms (e.g., multi-view consistency losses or self-refinement) to improve robustness?

3. How portable is the method without strong backbones (Qwen3-VL, SigLIP-2, Hunyuan3D-2.1)? Can smaller or alternative encoders maintain performance, and what adaptations are required?

**Limitations:**

yes

**Strengths And Weaknesses:**

Strengths

1. Unified framework: One backbone handles 3D understanding (caption/QA) and high-fidelity 3D generation, in an end to end way.

2. Mode-bound MoT: TokenAR (sequential) + BlockAR (parallel) with a hybrid causal and parallel mask enables efficient long-context reasoning, and fast 3D synthesis.

3. Direct 3D latent prediction: The MLLM outputs discrete 3D VAE tokens, which are decoded to geometry and materials, avoiding external diffusion backends.

4. Efficiency gains: about 3 times faster 3D generation at 4096 tokens via block-level prediction.

5. Solid engineering: Effective 3D position strategy (block-level pos, permutation-invariant within block), progressive training (from 512 to 4096 tokens), CFG, and rectified-flow timesteps.

6. Strong results vs MLLM-based baselines: Better p-FID/p-KID/CLIP-IQA+/MUSIQ, and semantic consistency than SAR3D, ShapeLLM-Omni. Clear qualitative improvements.

7. Transparent practices: Detailed trainin or inference configs and ablations (e.g., AdaLN instability, uniform point sampling works better).


Weaknesses

1. Fidelity ceiling: <4k 3D-VAE token budget limits fine details and textures; trailing top commercial 3D systems.

2. 3D understanding gap: Weaker than specialized 3D-LLMs on pure 3D caption or reasoning, partly due to the smaller 3D-language datasets.

3. Evaluation comparability: Some 3D captioning relies on rendered images rather than native 3D inputs, which makes cross-paper comparisons less direct.

4. Limited ablation depth: Core choices (block size, masking variants, 3D pos-enc design) could use more systematic curves/studies.

5. Dependence on strong backbones: Gains partly hinge on Qwen3-VL and SigLIP-2; portability to weaker or pretrained alternatives is uncertain.

6. Geometry/material trade-offs: VAE reconstruction and pipeline choices (e.g., mesh thickening) can smooth, or lose fine geometric features.

7. Scaling risks: Extending beyond 4k tokens, and adding richer materials may challenge memory, training stability, and latency without further architectural changes.

---

> ### Author Rebuttal · Authors · 2026-03-31
>
> We sincerely thank Reviewer WZkr for the comprehensive and insightful review. The detailed analysis of both strengths and limitations, along with the specific questions on scaling, robustness, and portability, has been extremely valuable for improving our work.
>
> **Q1: Fidelity ceiling (<4K tokens) and Scaling risks.**
>
> A1:We acknowledge this. However, we note that LLM context lengths are scaling rapidly: multiple recent LLMs already support 1M+ context windows, and a growing body of work proposes efficient long-context acceleration. We therefore believe the current token-budget constraint is not fundamental and can be expanded to much larger budgets in the future. For our model, we choose an economical context length that balances fidelity with practicality under our computational resource constraints. We will discuss token-budget scaling strategies in the revised manuscript.
>
> **Q2:3D caption.**
>
> In 3D caption generation, our results show that our model outperforms Qwen3-VL-2B prior to our training enhancements, and also surpasses multiple 7B/13B vision-language models. This suggests that our training procedure enhances the LLM’s spatial understanding; specifically, by explicitly introducing 3D-modality supervision and training, our method strengthens the LLM’s spatial perception and reasoning capability.
>
> **Q3: Limited ablation/Block size and token budget scaling curves.**
>
> A3: We will include speed and memory curves against token budget, covering the range from 512 to 5120 tokens in revision.
>
> Table 1 reports the training resource usage under 8 GPUs, with 20,480 tokens per GPU. Because the total token budget is fixed, our memory usage decreases as the number of 3D tokens increases, since this reduces the total number of training samples—i.e., the number of samples processed by the multimodal encoders.
> Table 2 summarizes how inference time scales with 3D-token length: memory grows linearly, while runtime scales quadratically with a small leading constant.
>
> However, regarding quality curves, changing the block size requires retraining the model from scratch. Therefore, as shown in Table 3, we currently report ablations at 512 and 4096 tokens only. The results show that increasing the token budget from 512 to 4096 yields consistent and substantial gains.In addition, the Table 3 include further ablation studies to clarify the contribution of each component.
>
> **Q4:VAE trade-offs.**
>
> A4:Our architecture is not tightly coupled to a specific spatial VAE. Our results indicate that the LLM can generate latents with strong 3D spatial structure, and the VAE serves as a swappable module in our framework. Therefore, if a VAE with better reconstruction fidelity and a reasonable token budget becomes available, it can be integrated with minimal changes by replacing this component.
>
> **Q5: Failure modes with ambiguous/noisy inputs.**
>
> A5: When conditioning on ambiguous or noisy images, typical failure modes include incomplete geometry, incorrect part relations, and reduced structural fidelity; for overly long instructions, failures often arise from diluted or conflicting constraints. A practical and widely adopted solution is to first transform out-of-domain inputs into more in-domain ones, e.g., by applying foreground extraction to noisy images, super-resolution or dehazing to blurry images, and context compression to long instructions.
>
> **Q6: Portability without strong backbones.**
>
> A6:In Table 3, we provide an additional ablation study, which shows that our framework can also achieve strong scores with a Qwen2.5-0.5B backbone. Notably, in this setting, Qwen2.5-0.5B is not pre-aligned as a vision-language model, and we initialize the vision encoder from siglip-so400m-14-980-flash-attn2-navit, rather than using a vision encoder that has already been aligned with the LLM. This demonstrates that our method does not rely on an already aligned VLM or a specific multimodal encoder; instead, it allows flexible swapping and mixing of backbones and multimodal encoders, while still achieving good performance even with a weaker backbone.
>
>
> **Table 1.** Training cost.
> |token|512|1024|2048|4096|5120|
> |:----|:----|:----|:----|:----|:----|
> |step/s|0.26|0.27|0.27|0.22|0.2|
> |memory|75G|60G|54G|43G|40G|
> |total samples|300|230|150|100|90|
> |3d generation samples|118|78|50|32|24|
>
> **Table 2.** Inference cost.
> |token|512|1024|2048|4096|5120|
> |:----|:----|:----|:----|:----|:----|
> |time|12s|13s|19s|35s|45s|
> |memory|9G|9G|10G|11G|12G|
>
>
> **Table 3.** Ablation study.
> |Hunyuan3D-2.1 VAE|MOT|LLM Backbone|token_num|p-FID(↓)|p-KID(↓)|
> |:----|:----|:----|:----|:----|:----|
> |x|x|Qwen2.5-0.5B|512|53.66|1.76|
> |o|x|Qwen2.5-0.5B|512|44.91|1.42|
> |o|o|Qwen2.5-0.5B|512|30.6|0.77|
> |o|o|Qwen3VL-2B|512|15.61|0.43|
> |o|o|Qwen2.5-0.5B|4096|16.57|0.53|
> |o|o|Qwen3VL-2B|4096|12.55|0.27|

---

> > ### Author Rebuttal · Reviewer_WZkr · 2026-04-03
> >
> > Thanks the authors for the responses in detail. My concerns are addressed with evidences, i.e., the training/inference cost and more ablation studies. I tend to accept this submission. While I'm also open to further discussion with other reviewers.

---

> > > ### Author Response · Authors · 2026-04-04
> > >
> > > Many thanks for your positive comments.

---

### Official Review · Reviewer_Sd2r · 2026-03-12

**Soundness:** 2
**Presentation:** 2
**Significance:** 2
**Originality:** 2
**Overall Recommendation:** 4
**Confidence:** 4

**Summary:**

This paper proposes CG-MLLM, a multimodal large language model that unifies 3D captioning and 3D generation within a single end-to-end framework. The core architectural contribution is a Mixture-of-Transformers design that binds a TokenAR Transformer and a BlockAR Transformer to a shared attention mechanism, initialized from Qwen3-VL weights. 3D content is encoded and decoded via a frozen Hunyuan3D2.1-VAE. The model is evaluated on generation metrics and captioning metrics, showing improvements over prior MLLM-based 3D generation methods.

**Compliance With Llm Reviewing Policy:**

Affirmed.

**Final Justification:**

solved all my problems, raise my score to 4

**Key Questions For Authors:**

1. The quantitative comparison in Table 1 includes only two baselines. Could the authors provide comparisons against recent non-LLM-based 3D generation methods using the same evaluation metrics? Without this, the scope of the "state-of-the-art" claim is unclear.
2. Could the authors provide an ablation isolating the contribution of the BlockAR MoT extension specifically—for example, by comparing against a baseline that uses the same Hunyuan3D VAE with a standard diffusion decoder, to quantify what the LLM integration adds?

**Limitations:**

yes

**Strengths And Weaknesses:**

Strengths:
1. The task framing is a meaningful step toward unified multimodal intelligence. The mode-binding rather than task-binding distinction from prior MoT methods is a technically interesting design choice.
2. The paper is transparent about failure modes and limitations, including the VAE reconstruction artifacts, AdaLN instability, and the performance gap versus leading commercial systems.
3. The 3×inference speedup from BlockAR parallel processing over token-level processing at 4096 token resolution is a practically meaningful result.

Weakness:
1. Table 1 compares only against SAR3D and ShapeLLM-Omni. Non-LLM-based 3D generation methods are entirely absent from quantitative comparison. The claim of state-of-the-art is only valid within the narrow MLLM-based 3D generation category, which contains very few methods.
2. Both the Spatial-VAE (Hunyuan3D2.1) and the VLM backbone (Qwen3-VL) are largely frozen or directly inherited. The novel trainable components are the MLP-Connector, the Embedding Layer, and the BlockAR extension. The paper does not sufficiently ablate how much of the generation quality stems from the pretrained VAE quality versus the proposed architecture.
3. All qualitative and quantitative results are on single isolated objects from Objaverse-style datasets. There is no evaluation on scene-level 3D generation or text-conditioned generation on out-of-distribution prompts, leaving the claim of compositional 3D spatial understanding largely unsubstantiated.

---

> ### Author Rebuttal · Authors · 2026-03-31
>
> We sincerely thank Reviewer Sd2r for the time and efforts in reviewing our work. We appreciate the recognition of our mode-binding design philosophy, and the constructive questions have helped us identify directions for strengthening the paper.
>
> **Q1: Limited baselines in Table 1.**
>
> A1: We have added more comparison with SoTA diffusion-based 3D generation methods, and will include it in revision. CG-MLLM achieves generation quality comparable to dedicated diffusion-based 3D generation systems that typically employ significantly far more compute optimized exclusively for generation.
>
> **Table 1.** Comparison.
> |Method|p-FID(↓)|p-KID(↓)|clipiqa+(↑)|musiq(↑)|clip(↑)|user-study(↑)|
> |:----|:----|:----|:----|:----|:----|:----|
> |Diffusion-Base| | | | | | |
> |michelangelo|17.96|0.56|0.45|71.42|84.08|2.6|
> |craftsman|14.09|0.4|0.45|71.09|**84.86**|3.15| | | |
> |hunyuan3d-2.1|16.8|0.53|**0.47**|71.2|**85.11**|3.15|
> |trellis|**7.36**|**0.12**|0.44|66.97|84.13|3.28|
> |sam3d|33.92|1.13|**0.47**|70.21|**84.67**|**3.45**|
> |MLLM-Base| | | | | | |
> |sar3d|30.07|1|0.42|66.01|82.86|2.93|
> |shapellm-omni|13.11|0.29|0.37|55.71|84.18|2.3|
> |ours|**12.55**|**0.27**|**0.45**|**71.65**|**84.47**|**3.32**|
>
> **Q2: Ablation of Architectural Gains.**
>
> A2: We emphasize that our end-to-end architecture requires careful training adjustments and is not a simple concatenation of an LLM and a diffusion head. To better disentangle the source of the gains, we will add a new VAE ablation and clarify an existing controlled comparison in the revision.
> (1) VAE ablation. We will replace the Hunyuan3D-2.1 VAE with the Hunyuan3D-2mini VAE to disentangle the effect of VAE quality from that of our proposed architecture.
> (2) BlockAR vs. standard DiT. This is an existing experiment rather than a new one. As shown in Fig. 6(a), our LLM-integrated BlockAR outperforms a fully converted standard DiT variant, indicating that retaining LLM characteristics is important. We will make this result more explicit in the revision. Details are discussed in Q4 below.
> (3) Backbone scaling ablation. By instantiating our framework on Qwen2.5-0.5B and comparing it with the 2B model, we show that the proposed method is effective even on a small backbone and exhibits a favorable scaling trend as model size increases. Since Qwen2.5-0.5B does not include a pretrained vision encoder, the visual extractor in this variant is initialized from SigLIP and trained accordingly.
>
> **Table 2.** Ablation studies.
> |Hunyuan3D-2.1 VAE|MOT|LLM Backbone|token_num|p-FID(↓)|p-KID(↓)|
> |:----|:----|:----|:----|:----|:----|
> |x|x|Qwen2.5-0.5B|512|53.66|1.76|
> |o|x|Qwen2.5-0.5B|512|44.91|1.42|
> |o|o|Qwen2.5-0.5B|512|30.6|0.77|
> |o|o|Qwen3VL-2B|512|15.61|0.43|
> |o|o|Qwen2.5-0.5B|4096|16.57|0.53|
> |o|o|Qwen3VL-2B|4096|12.55|0.27|
>
> **Q3: Scene-level evaluation.**
>
> A3: Existing 3D scene generation methods generally fall into two paradigms. The first directly generates entire scenes through neural networks, but struggles to balance fine-grained detail and scene scale simultaneously. The second adopts a layout-object decoupled pipeline, where a layout planner determines spatial arrangement and individual objects are generated separately, allowing both detail quality and scene coverage to be maintained. Our method is naturally suited for the second paradigm, as CG-MLLM can serve as the object generator within such a pipeline, and its 3D understanding capability can further assist layout reasoning. Moreover, we argue that spatial understanding should also include the understanding of complex object topology, occlusion, and part-level geometric relations, which our model already exhibits strongly.
>
> **Q4: Ablation isolating BlockAR MoT contribution.**
>
> A4: Unlike TokenAR, BlockAR applies bidirectional attention among tokens within a block, and the tokens in a block share a single positional encoding in the LLM, so the whole block is treated as one joint unit during both training and inference—consistent with the typical DiT formulation. Consequently, the primary remaining difference between BlockAR and a standard diffusion DiT lies in how layer normalization is implemented. We describe this ablation in Sec. 4.3 and Fig. 6(a): specifically, we replace BlockAR’s layer normalization with standard AdaLN (conditioned on the diffusion timestep). However, such a modification results in substantially higher training loss. This demonstrates that retaining the LLM's normalization scheme is beneficial. We hypothesize that preserving the original LLM structure keeps the semantic space of the generation branch better aligned with the understanding branch, enabling stronger joint performance. We will clarify and re-explain this setup in the revision.

---

> > ### Author Rebuttal · Reviewer_Sd2r · 2026-04-02
> >
> > Here's a more concise version:
> >
> > ---
> >
> > **Rebuttal Acknowledgement by Reviewer Sd2r**
> >
> > **Acknowledgement:** (b) Partially resolved - I have follow-up questions for the authors.
> >
> > **Reasons:**
> >
> > I thank the authors for the detailed rebuttal and additional experiments. Below I address each point.
> >
> > Q1 (Expanded baselines): The inclusion of diffusion-based methods is appreciated. However, the results also show that CG-MLLM substantially trails Trellis (p-FID 7.36 vs. 12.55) and that SAM3D achieves a higher user study score despite worse automated metrics. I encourage the authors to (1) include these comparisons in the main paper with an honest discussion of the performance gap, and (2) provide qualitative comparisons against diffusion baselines, as 3D generation metrics alone can be unreliable.
> >
> > Q2 (Ablation): The new ablation table effectively decomposes the contributions of VAE, MoT, and backbone scale — this largely addresses my concern. One minor clarification: what generation architecture is used in the non-MoT rows? A standard DiT head or a token-level AR decoder?
> >
> > Summary: The rebuttal has improved my understanding and the ablation table is a solid addition. Given the interesting architectural ideas and improved evidence, I am raising my score from 3 to 4. However, the missing qualitative comparisons with diffusion baselines, the performance gap with Trellis, and the lack of scene-level evaluation remain concerns that should be addressed in the revision.

---

> > > ### Author Response · Authors · 2026-04-04
> > >
> > > We thank Reviewer Sd2r for raising the score from 3 to 4. In the revision, we will include qualitative comparisons with non-MLLM-based baselines, scene-level evaluations, and a discussion of the performance gap with non-MLLM methods.
> > >
> > > Q1 non-MLLM-based methods
> > >
> > > In the revision, we will include the results presented in the current rebuttal phase, including comparisons with non-MLLM-based methods and ablation experiments, and add discussion of the performance. We provide an expanded version of the original Fig. 4  at https://anonymous.4open.science/r/260404_tmp-CD4E/qualitative_comparisons.png . We will also present additional qualitative experiments in the revision.
> > >
> > > Q2  What generation architecture is used in the non-MoT rows
> > >
> > > Our method consists of two Transformers: TokenAR handles token-level task, while BlockAR handles block-level task. In contrast, the non-MoT ablation uses only one Transformer to jointly process both token-level and block-level signals. This baseline does not introduce an additional diffusion head, but it is not a vanilla token-level autoregressive decoder either. It is obtained by extending a token-level AR decoder and modifying the causal attention mask to enable partial parallelism during the forward pass.
> > >
> > >
> > >
> > >
> > > Thank you again for your detailed feedback and for recognizing the improvements in our rebuttal. We are pleased that the additional ablation studies helped clarify our approach.
> > >
> > > **We note that you mentioned raising your score from 3 to 4, and we sincerely appreciate your consideration. We would be grateful if you could let us know whether you have had an opportunity to update your score and submit the final justification.**

---

### Official Review · Reviewer_o9TK · 2026-03-12

**Soundness:** 2
**Presentation:** 3
**Significance:** 2
**Originality:** 2
**Overall Recommendation:** 4
**Confidence:** 4

**Summary:**

This paper proposes a multi-modal (text, image, and 3D) language model for 3D generation and understanding within a unified framework. This is beneficial because, in this way, large language models which are strong at understanding and generating text can natively and effectively generate 3D objects without requiring any external model. It also enables the 3D domain to benefit more directly from the rapid progress and scaling momentum of LLMs.

To approach this problem, the authors propose a supervised training pipeline composed of two interleaved transformer models. The two main transformer modules are:
- TokenAR, which maintains the model’s original capacity for token-level autoregressive modeling
- BlockAR, which extends this design to handle parallel block tokens.

This design provides a unified framework that allows the pipeline to leverage the knowledge of the vision language models for 2D visual and textual understanding while also empowering it with robust spatial perception and 3D generative capabilities. The method is evaluated against several baselines both quantitatively and qualitatively.

**Compliance With Llm Reviewing Policy:**

Affirmed.

**Final Justification:**

I believe the paper presents interesting methodological contributions and the authors have made a solid effort to address my concerns. However, the work remains limited in terms of performance gap with native 3D baselines which is also confirmed by the authors. Therefore, this limitation along with both qualitative and quantitative results, should be clearly and explicitly highlighted in the paper. Without such discussion, the presentation gives a misleading impression of the method’s strength relative to existing state-of-the-art 3D native approaches.

But, given the strengths of the methodology and other aspects of the paper, I am increasing my score to 4 (weak accept). I would, however, also support a rejection if the requested modifications and other clarifications are not provided, or if the overall reviewer consensus leans in that direction.

**Key Questions For Authors:**

- In Table 1, it is not clear how many samples (3D objects) were used to report these results, nor from which dataset these objects were selected. Additionally, it is unclear whether the evaluation samples were drawn from the same dataset used for training or from a separate dataset.

- What is the order of magnitude for the number of vertices and faces in the meshes that the model can generate? For example, is the model capable of generating meshes with triangular faces on the order of one million faces, or at least around 100k faces?

- Is the LLM able to generate only the geometry of the mesh, or can it generate both geometry and appearance (e.g., textures) as well? If the model generates only the geometry, how are the appearances of the meshes in the results (e.g., Figure 4) produced?

**Limitations:**

- Although the paper presents results on various shapes, the method’s effectiveness on highly intricate or challenging 3D geometries, particularly those with many occluded or self-intersecting parts, is not fully explored. Therefore, the examples provided in the paper may not represent the full spectrum of complex 3D models.

- Since the scope of baseline comparisons is limited, it is difficult to assess how strong and effective this method is for the 3D generation task.

**Strengths And Weaknesses:**

**Strengths:**

- The paper is easy to follow and addresses a meaningful problem in the multi-modal 3D generation and understanding literature. The framework transitions smoothly from the standard multi-modal generation to its specific contributions and architectural changes. The literature review and related work are thorough and well written.

- The way the paper combines two pre-trained vision–language transformer models and integrates them for 3D generation and understanding is promising. This design choice provides a unified framework that allows the pipeline to leverage the knowledge of VLMs for 2D visual and textual understanding while also empowering it with robust spatial perception and 3D generative capabilities.


**Weaknesses:**

- The work appears somewhat incremental, as most of its components are derived from previously proposed methods.

- There is a lack of sufficient baseline comparisons. For example, why do the authors not compare their quantitative and qualitative results with native 3D generation models such as TRELLIS [1], SAM3D [2], etc.? I understand that this work is an MLLM-based 3D generation approach, which is inherently different from native 3D generation methods. However, the ultimate goal of both approaches is the same: generating visually pleasing 3D objects. Therefore, it would be reasonable to compare the 3D outputs produced by this method with those generated by native approaches. Such a comparison could reveal whether the proposed method still lags behind native methods, outperforms them, or performs at a comparable level. In any case, this comparison would help readers and the community better understand where MLLM-based approaches lie within the broader 3D generation spectrum and provide clearer insight into how the proposed method fits within the overall 3D generation research landscape.

- There is no visualization of ground-truth 3D objects in the qualitative results (e.g. Figure 4). Since the training approach is based on supervised learning with ground truth, the qualitative results should include comparisons with the ground-truth objects. This would give readers a clearer idea of how well the model performs independently of comparisons with other methods.

- There is no user preference study. In the 3D generation literature, the perception of a “good” 3D object is often subjective and can vary significantly between individuals. Given this, it would be beneficial to include a user study where participants evaluate the quality of the generated 3D objects.

- There is a lack of component analysis. Ablation studies examining individual contributions would help readers understand which architectural changes or components lead to the observed improvements (for example, different strategies for the transformer specialization logic that alternates between TokenAR and BlockAR).

- There is no report of computational cost for either inference or training. Since the task involves fine-tuning two transformer models with supervision, the training process could be relatively computationally intensive. However, the paper does not report the computational requirements for training or inference, nor the memory requirements needed to generate and understand 3D objects. Providing this information would help practitioners better understand the compute resources required to apply this method.




**References:**

[1] Xiang, Jianfeng, et al. "Structured 3D Latents for Scalable and Versatile 3D Generation." Proceedings of the IEEE/CVF Conference on Computer Vision and Pattern Recognition. 2025.

[2] Chen, Xingyu, et al. "Sam 3D: 3Dfy Anything in Images." arXiv preprint arXiv:2511.16624 (2025).

---

> ### Author Rebuttal · Authors · 2026-03-31
>
> We sincerely thank Reviewer o9TK for the considerable time and efforts dedicated to reviewing our paper.
>
> **Q1: Incremental contribution.**
>
> A1: Our key contributions are: (1) differentiating TokenAR and BlockAR at the *mode* level rather than the *task* level, a fundamentally different design philosophy from prior MoT methods; (2) a novel unified approach for image and 3D understanding beyond simple multi-task learning;  (3) a MLLM that unifies text understanding and generation, image understanding, 3D understanding, and 3D generation within a single compact 2B-MoT model. Our model achieves SOTA in 3D generation and substantially improves 3D-aware understanding from images, outperforming several much larger 7B and 13B models. These results demonstrate architectural efficiency rather than brute-force scaling.
>
> **Q2: Comparisons with native 3D generation methods.**
>
> A2: We  have conducted these comparisons. Please refer to Table 1 for the results. We acknowledge a fidelity gap with the strongest native methods, but note that our method achieves competitive performance to SOTA native 3D generation models such as HunYuan3D-2.1 and TRELLIS. We attribute the remaining gap with top systems to differences in computation and model scale rather than fundamental limitations.
>
> **Q3: Ground-truth visualization.**
>
> A3:We will include the ground-truth visualization in revision to facilitate visual comparison.
>
> **Q4: User study.**
>
> A4: As shown in Table 1, we conducted a user study and will include the full results in the revision. Overall, we collected 100 questionnaires, resulting in approximately 2,400 data points for a statistically meaningful analysis of user preference. In the user study, we rank second only to SAM3D.  Overall, comparisons show that CG-MLLM achieves comparable generation quality to dedicated diffusion-based 3D generators that typically require substantially greater computational cost .
>
> **Q5: Ablation studies.**
>
> A5: In response to this concern, we have conducted additional ablations and a more fine-grained analysis, and will include them in the revision. Specifically, these include: (1) a VAE variant ablation; (2) an ablation that removes MoT and replaces it with a single-stream LLM; (3) an LLM backbone scaling ablation from 0.5B to 2B, demonstrating that our method follows a consistent scaling trend; (4) a clearer comparison between BlockAR and a standard DiT design; and (5) a block-size/token-budget scaling study with speed and memory curves from 512 to 8192 tokens.
>
> **Q6: Computational cost.**
>
> A6: All experiments were conducted with our model on 16 H20 GPUs, with a training time of approximately two weeks. Table 2 reports the inference time and memory footprint under different token budgets. We will include it in the revision.
>
> **Q7: Evaluation dataset and sample count.**
>
> A7: We evaluate on the Toys4K dataset (not included in training set), using 3225 samples. We will clarify this.
>
> **Q8: Vertex/face count/self-occlusion/self-intersecting.**
>
> A8: Our method generates SDFs, so there is no inherent face-count limit. In practice, we use a 256-resolution octree for SDF extraction, yielding approximately 100K faces, and then apply decimation to reduce the mesh to under 40K faces. Fig. 10, Line 3 of the paper provides an example of self-occlusion and visually overlapping structures, where the leftmost panel shows the input image and the right panels present the generated results. The input exhibits severe self-occlusion and highly overlapping grid-like structures in the chair backrest, making the spatial layout particularly challenging to infer. Nevertheless, our method successfully reconstructs the complete iron-frame structure and preserves the correct spatial relationships among the backrest, seat, and multi-legged base.
>
> **Q9: Geometry/appearance.**
>
> A9: As described in Sec. 3.3, Textures are produced by Hunyuan3D's texture generator as post-processing. Joint geometry-appearance generation remains an active research topic. Although joint generation has been explored, generating geometry and appearance separately remains the most widely adopted practice in both academic and commercial systems. We will clarify this more explicitly in revision.
>
> **Table 1.** Comparisons.
> |Method|p-FID(↓)|p-KID(↓)|clipiqa+(↑)|musiq(↑)|clip(↑)|user-study(↑)|
> |:----|:----|:----|:----|:----|:----|:----|
> |Diffusion-Base| | | | | | |
> |michelangelo|17.96|0.56|0.45|71.42|84.08|2.6|
> |craftsman|14.09|0.4|0.45|71.09|**84.86**|3.15| | | |
> |hunyuan3d-2.1|16.8|0.53|**0.47**|71.2|**85.11**|3.15|
> |trellis|**7.36**|**0.12**|0.44|66.97|84.13|3.28|
> |sam3d|33.92|1.13|**0.47**|70.21|**84.67**|**3.45**|
> |MLLM-Base| | | | | | |
> |sar3d|30.07|1|0.42|66.01|82.86|2.93|
> |shapellm-omni|13.11|0.29|0.37|55.71|84.18|2.3|
> |ours|**12.55**|**0.27**|**0.45**|**71.65**|**84.47**|**3.32**|
>
> **Table 2.** Inference cost.
> |token|512|1024|2048|4096|5120|
> |:----|:----|:----|:----|:----|:----|
> |time|12s|13s|19s|35s|45s|
> |memory|9G|9G|10G|11G|12G|

---

> > ### Author Rebuttal · Reviewer_o9TK · 2026-04-02
> >
> > I appreciate the authors’ responses and their efforts to include additional results, such as comparisons with native 3D baselines, a user study, and ablation analyses. I also acknowledge that the authors recognize the performance gap with native 3D methods. Since most of my concerns have been addressed, I am raising my score from 3 (weak reject) to 4 (weak accept). However, the scope of the work still feels limited due to several remaining concerns:
> >
> > - Although the authors provide Table 1 in the rebuttal to quantitatively compare against native 3D baselines (which I appreciate), in 3D literature, quantitative metrics alone are often insufficient and may not fully support qualitative claims. Therefore, the paper would benefit from substantially more qualitative comparisons to better validate the results in Table 1 and support the claim of outperforming or being competitive with prior native 3D methods.
> >
> > - From Table 1 and authors' explanation, the evaluation setup is not clearly described (e.g., which dataset was used). I assume it is the Toys4K dataset mentioned by the authors, but this should be clarified. More importantly, it is somewhat surprising to see SAM3D underperforming TRELLIS by a large margin on the first two metrics (p-FID and p-KID). This is unexpected given that SAM3D is a more recent method and reports strong improvements over TRELLIS in its own paper. This discrepancy likely stems from differences in evaluation setups (datasets and metrics). I recommend that the authors report results using the same benchmarks as SAM3D (e.g., SA-3DAO and the ISO3D Eval Set from Table 2 of the SAM3D paper) to provide a clearer and more consistent comparison.
> >
> > - Even under the current evaluation setup in Table 1 here, the results suggest that the proposed method does not outperform state-of-the-art native 3D approaches such as SAM3D, TRELLIS, and HunYuan3D-2.1. This indicates that MLLM-based methods still have room for improvement. I strongly encourage the authors to include these comparative results along with the qualitative comparisons mentioned above and to clearly discuss the limitations and performance gap relative to native 3D methods in the main paper.
> >
> > - The authors attribute the performance gap between their MLLM-based method and native 3D approaches to differences in computation and model scale. However, the evidence provided does not fully support this claim. For instance, in TRELLIS, the largest model (TRELLIS-text-xlarge) has around 2B parameters, which is comparable in scale to the 2B-parameter model used in this work. The authors should clarify this point and provide supporting evidence. If such evidence is not available, it would be more appropriate to frame this as an open question or direction for future work, rather than asserting a specific cause without justification.
> >
> > **Final remarks for the AC and authors:** I believe the paper presents interesting methodological contributions, and the authors have made a solid effort to address my concerns. However, the work remains limited in terms of qualitative comparisons with native 3D baselines, and the performance gap with these methods which is acknowledged by the authors. Therefore, it still needs clearer analysis and discussion. More extensive qualitative evidence is necessary to support the quantitative findings in Table 1 presented here in the rebuttal phase. But, given the strengths of the methodology and other aspects of the paper, I am increasing my score to 4 (weak accept). I would, however, also support a rejection if the requested qualitative evidence and other clarifications are not provided, or if the overall reviewer consensus leans in that direction.

---

> > > ### Author Response · Authors · 2026-04-04
> > >
> > > We thank Reviewer o9TK for recognizing that our paper presents interesting methodological contributions and for increasing the score to 4 (weak accept). We will revise the paper to provide a more detailed discussion of the performance gap, framing the relative contributions of parameter allocation, training paradigm, and architectural choices as important directions for future investigation, rather than attributing the gap to a single cause. We also acknowledge that a gap remains with the strongest non-MLLM-based methods, such as SAM3D, and that closing this gap while preserving the unified generation-understanding capability remains a key goal for future work. Below, we provide responses to some remaining questions.
> > >
> > > Q1 qualitative comparisons
> > >
> > > We provide an expanded version of the original Fig. 4 at https://anonymous.4open.science/r/260404_tmp-CD4E/qualitative_comparisons.png. We will also present additional qualitative experiments in the revision.
> > >
> > > Q2 evaluation setup
> > >
> > > Yes, we conducted the comparisons using Toys4K. The explanation for the first two metrics of SAM3D is as follows: Since the officially released SAM3D demo generates 3DGS results and provide the accompanied point clouds extracted from 3DGS centers. The feature distribution computed from these point clouds might be different from the point-clouds sampled from the meshes in Toys4K. We speculate that such a difference is the main reason for the relatively poor performance on the p-FID and p-KID metric values of SAM3D. We also discuss this point in the reply to reviewer “Z2gQ ”.
> > >
> > > Additionally, to the best of our knowledge, the SA-3DAO dataset is not publicly available, and the ISO3D dataset only contains images without ground-truth geometry, so we are currently unable to supplement experiments on these two datasets.
> > >
> > > Q3 performance gap with native 3D methods as an open problem
> > >
> > > We apologize the confusion caused in our rebuttal. Our CG-MLLM has approximately 2B dynamically activated parameters. Its parameter scale is comparable to TRELLIS-text-xlarge. While our BlockAR is smaller than some models, such as Hunyuan3D-2.1 (3.3B), attributing the observed performance gap solely to parameter scale is overly simplistic.
> > >
> > > **We agree that the performance gap between MLLM-based and native 3D generation methods should be treated as an open question, and will discuss this limitation clearly in the revision.**
> > >
> > > Q4: Future directions and the potential of MLLM
> > >
> > > As pointed out in reviewer’s comments, there are still rooms to improve the performance of MLLM in 3D generation. We would like to share a few thoughts as follows:
> > >
> > > 1.Stronger multi-encoder image conditioning. Methods such as Trellis and Hunyuan3D-2.1 inject image conditions via DINO-style encoders, whereas we currently rely on a CLIP-style encoder. Empirically, CLIP-style features tend to capture high-level semantics well but can lose fine-grained visual detail. In future work, combining multiple complementary encoders (e.g., DINO + CLIP) could better jointly satisfy semantic alignment and fine-grained perception. This idea is already explored in some MLLMs that jointly use DINO-based and CLIP-based encoders, and a similar strategy could be incorporated into our framework.
> > >
> > > 2.Scaling up model parameters. Recent work suggests that both LLMs and 3D generation follow scaling laws; we therefore expect that increasing model size can materially improve performance. For scaling up, MLLM-based approaches can be more training-efficient in practice because they can build on larger pretrained LLMs, inherit existing knowledge, and extend it with additional training, whereas many non-MLLM pipelines must be trained largely from scratch.
> > > Rough training-computation sketch (for reference). For our method, a coarse accounting is
> > > 200k steps × 16 GPUs × ~40k tokens × ~2B activated parameters.
> > > For TRELLIS, we use the setup reported in the original paper:
> > > 400k steps × 64 GPUs × 4 samples × ~20k tokens × ~2B activated parameters.
> > > Under this (admittedly simplified) accounting, TRELLIS’s training computation is roughly 16× higher than ours, which highlights a potential efficiency advantage of MLLM-based scaling when pursuing larger models in future work.
> > >
> > > 3.Increasing the 3D token budget. In our current setup, we adopt a relatively conservative configuration and keep the per-sample context window around 4k tokens. However, LLM context lengths are advancing rapidly, and several models already support ~1M-token contexts. In future work, more efficient long-context training/inference could allow us to expand the context window and allocate more 3D tokens, which—together with a higher-fidelity reconstruction VAE at a reasonable token cost—is expected to further improve generation quality.

---

### Official Review · Reviewer_Z2gQ · 2026-03-13

**Soundness:** 3
**Presentation:** 3
**Significance:** 3
**Originality:** 3
**Overall Recommendation:** 4
**Confidence:** 3

**Summary:**

The authors focus on the multimodal 3D generation and understanding task. Specifically, they proposed a unified framework for joint 3D generation and 3D captioning. The proposed framework is a mixture-of-transformer architecture, including token-level autoregression and block-level autoregression. The experimental results show that CG-MLLM outperforms existing 3D generation MLLM methods. The quantitative comparison also shows that the 3D captioning metrics outperform existing methods.

**Compliance With Llm Reviewing Policy:**

Affirmed.

**Final Justification:**

The additional experimental results and the explanation have addressed my concerns. I will keep my overall recommendation as "weak accept".

**Key Questions For Authors:**

My key question is listed in weakness-1: How is the performance gap between CG-MLLM and current SOTA 3D generation methods?

**Limitations:**

yes.

**Strengths And Weaknesses:**

Strength:
1. The authors proposed a unified framework that supports both 3D generation and captioning. The novel design of the mixture-of-transformers is not a simple task-based approach, like one for generation and the other for understanding. In contrast, the proposed CG-MLLM involves token-level autoregression and block-level autoregression, which allows long-context interaction. From my understanding, it is a key contribution to the community.
2. The quantitative results and qualitative results both prove that the 3D generation capability outperforms current MLLM-based methods, and the 3D captioning also surpasses all the baselines.

Weakness:
1. The paper lacks the comparison with SOTA 3D generation methods, like TRELLIS-text/TRELLIS-image. Although the proposed method is MLLM-based, comparing with SOTA 3D generation methods without an MLLM architecture could help us to understand the capability gap between different technical paradigms.
2. Although Tables 1 and 2 show the strong performance of the proposed novel architecture, it seems that the ablation to prove the effectiveness of the designed modules is not enough. More ablation studies could help us understand the importance of different modules.

---

> ### Author Rebuttal · Authors · 2026-03-31
>
> We sincerely thank Reviewer Z2gQ for the time and efforts spent on our manuscript.
>
> **Q1: Comparison with SOTA non-MLLM 3D generation methods.**
>
> A1: We have now conducted additional comparisons with state-of-the-art diffusion-based 3D generation methods and will incorporate these results into the revised manuscript. As shown in Table 1, we systematically compare our method with representative generation-only diffusion-based SoTA methods across Six key dimensions. Top-tier results (the best and near-best scores) are highlighted in **bold**. We note that SAM3D uses point clouds extracted from 3DGS centers rather than mesh-sampled GT point clouds, so the distributions are mismatched. Therefore,  SAM3D’s FID/KID may be biased relative to mesh-sampled GT.
>
> As expected, on geometric quality metrics (p-FID and p-KID), our method surpasses methods such as Craftsman and Hunyuan3D-2.1, benefiting from the strong encoding-decoding capability of the Hunyuan3D-2.1 VAE. On rendered-image metrics, we comprehensively surpass Michelangelo and remain highly competitive compared to other methods. In the user study, we rank second only to SAM3D.  Overall, these comparisons show that CG-MLLM achieves comparable generation quality to dedicated diffusion-based 3D generators that typically require substantially greater computational cost, while simultaneously supporting text, image, and 3D understanding within a single unified framework, a capability absent from all compared methods.
>
> In addition, our newly defined paradigm is loosely coupled and flexible, making it possible in the future to be compatible with diffusion-based language models (a direction that has been actively explored in recent work). Leveraging the rapid development of MLLMs, we see a broader outlook for 3D generation and understanding.
>
> **Table 1.** Comparison.
> |Method|p-FID(↓)|p-KID(↓)|clipiqa+(↑)|musiq(↑)|clip(↑)|user-study(↑)|
> |:----|:----|:----|:----|:----|:----|:----|
> |Diffusion-Base| | | | | | |
> |michelangelo|17.96|0.56|0.45|71.42|84.08|2.6|
> |craftsman|14.09|0.4|0.45|71.09|**84.86**|3.15| | | |
> |hunyuan3d-2.1|16.8|0.53|**0.47**|71.2|**85.11**|3.15|
> |trellis|**7.36**|**0.12**|0.44|66.97|84.13|3.28|
> |sam3d|33.92|1.13|**0.47**|70.21|**84.67**|**3.45**|
> |MLLM-Base| | | | | | |
> |sar3d|30.07|1|0.42|66.01|82.86|2.93|
> |shapellm-omni|13.11|0.29|0.37|55.71|84.18|2.3|
> |ours|**12.55**|**0.27**|**0.45**|**71.65**|**84.47**|**3.32**|
>
> **Q2: More ablation studies.**
>
> A2: We have conducted the following ablation studies, which will be included in the revision with detailed analysis:
>
> (1) VAE backbone:  Comparing generation quality between weaker VAE (Hunyuan3D-2mini) and stronger VAE (Hunyuan3D-2.1) to isolate the impact of VAE representation capacity on our architecture's performance.
>
> (2) MOT architecture: The results in Table 2 clearly demonstrate the importance of the MOT design. We will further clarify the design choices behind BlockAR (i.e., standard DiT vs. LLM) in Section 4.3.
>
> (3) LLM backbone: Replacing Qwen3-VL with Qwen2.5-0.5B to evaluate portability to smaller backbones.
>
> (4) Object token length: Evaluating how different OBJ token lengths affect generation performance under the same model size.
>
> (5) Block size / token budget scaling: Speed and memory curves from 512 to 5120 tokens will be provided.
>
> Results are reported in Tables 2-4. Table 2 shows the gains from stronger Hunyuan3D-2.1 VAE and the MOT architecture, and increased 3D token allocation. For the LLM backobone ablation, we adapt Qwen2.5-0.5B to our framework; compared with Qwen3-VL 2B, the main differences lie in the backbone and vision encoder capacities. This ablation suggests that our method follow scaling-law behavior, i.e., increasing the model size yields further improvements. Table 3 reports training resource usage under 8 GPUs with 20480 tokens per GPU. Because the total token budget is fixed, memory consumption decreases as the number of 3D tokens increases, since this reduces the total number of training samples,such as samples processed by the multimodal encoders. Table 4 summarizes how inference cost scales with 3D-token length: memory grows linearly, while runtime scales quadratically with a small leading constant.
>
>
> **Table 2.** Ablation studies.
> |Hunyuan3D-2.1 VAE|MOT|LLM Backbone|token_num|p-FID(↓)|p-KID(↓)|
> |:----|:----|:----|:----|:----|:----|
> |x|x|Qwen2.5-0.5B|512|53.66|1.76|
> |o|x|Qwen2.5-0.5B|512|44.91|1.42|
> |o|o|Qwen2.5-0.5B|512|30.6|0.77|
> |o|o|Qwen3VL-2B|512|15.61|0.43|
> |o|o|Qwen2.5-0.5B|4096|16.57|0.53|
> |o|o|Qwen3VL-2B|4096|12.55|0.27|
>
> **Table 3.** Training cost.
> |token|512|1024|2048|4096|5120|
> |:----|:----|:----|:----|:----|:----|
> |step/s|0.26|0.27|0.27|0.22|0.2|
> |memory|75G|60G|54G|43G|40G|
> |total sample|300|230|150|100|90|
> |3d generation sample|118|78|50|32|24|
>
> **Table 4.** Inference cost.
> |token|512|1024|2048|4096|5120|
> |:----|:----|:----|:----|:----|:----|
> |time|12s|13s|19s|35s|45s|
> |memory|9G|9G|10G|11G|12G|

---

> > ### Author Rebuttal · Reviewer_Z2gQ · 2026-04-04
> >
> > The additional experimental results and the explanation have addressed my concerns.

---

> > > ### Author Response · Authors · 2026-04-04
> > >
> > > Many thanks for your positive comments.

---

### Decision · Program_Chairs · 2026-04-30

**Decision:**

Accept (regular)

**Comment:**

All reviewers are consistently positive. The AE also read all comments and agree with all reviewers. It is a clear accept.